# Binarized Neural Machine Translation

**Yichi Zhang**[*]
Cornell University
yz2499@cornell.edu

**Ankush Garg**[*]
Google DeepMind
ankugarg@google.com

**Yuan Cao**
Google DeepMind
yuancao@google.com

**Łukasz Lew**
Google Research
lew@google.com

**Behrooz Ghorbani**[†]
OpenAI
ghorbani@openai.com

**Zhiru Zhang**
Cornell University
zhiruz@cornell.edu

**Orhan Firat**
Google DeepMind
orhanf@google.com

## Abstract

The rapid scaling of language models is motivating research using low-bitwidth quantization. In this work, we propose a novel binarization technique for Transformers applied to machine translation (BMT), the first of its kind. We identify and address the problem of inflated dot-product variance when using one-bit weights and activations. Specifically, BMT leverages additional LayerNorms and residual connections to improve binarization quality. Experiments on the WMT dataset show that a one-bit weight-only Transformer can achieve the same quality as a float one, while being $16\times$ smaller in size. One-bit activations incur varying degrees of quality drop, but mitigated by the proposed architectural changes. We further conduct a scaling law study using production-scale translation datasets, which shows that one-bit weight Transformers scale and generalize well in both in-domain and out-of-domain settings[3].

## 1 Introduction

Neural language models are scaling, with the parameter count of recent models, such as the GPT family, roughly increased by $10\times$ per year [29]. A scaling law study by Kaplan et al. [21] suggests that the continuous increase in model parameters is strongly correlated with performance improvement. This trend has been validated by recent successes in large-scale models, such as the 540-billion parameter Pathways Language Model (PaLM), which achieves breakthrough performance on language understanding and generation [11]. The 540-billion parameter Minerva [25] also exceeded the national average on the National Math Exam in Poland in 2021, where language models were previously far from human-level. Similarly, in the field of neural machine translation (MT), the scaling law holds, as reported by Ghorbani et al. [16], with the translation quality improving as the model size increases.

The aggressive scaling trend resulted in unprecedented challenges in model serving. In particular:

**The inference cost grows exponentially.** The size and computational complexity of language models are increasing rapidly, with roughly a $10\times$ increase in model size and a $100\times$ increase in operation

---

[*]Equal contribution.

[†]Work done while at Google.

[3]Source code is available in the init2winit library: `https://github.com/google/init2winit/blob/master/init2winit/model_lib/xformer_translate_binary.py`

count per year [18]. However, the energy efficiency of hardware used to run these models is not keeping pace. Specifically, the energy required for FP32 operations has improved by only $2.5\times$ over the past 11 years (2007-2018), from 45nm to 7nm process nodes. Over the same period, DRAM access energy has only improved by $6.3\times$ [20]. The ever-growing gap between model size inflation and inefficiency in hardware energy utility is causing inference energy to grow exponentially, which is becoming a major cost of running language models in datacenters.

**The inter-chip communication overhead becomes non-negligible.** Data parallelism alone is no longer sufficient for models at such a large scale since one matrix multiplication cannot fit on a single accelerator chip. Each weight tensor in PaLM [11], for example, is partitioned across 3072 TPUv4 chips in a pod. This leads to a huge overhead on transferring the weights and intermediate activations across the datacenter networks.

**Latency-critical applications can now hardly benefit from parameter caching.** Loading model parameters from DRAM to on-chip accelerator memory often takes a lot of time during inference. In the past, parameter caching was an effective optimization for latency because it reused model weights and avoided off-chip memory transfers. However, evaluations on edge TPUs reported that this method works best for models with fewer than 30 million parameters [36]. For larger models, parameter caching even becomes harmful. Benefits from compiler optimizations are diminishing, and the serving latency becomes almost proportional to the model parameter count. In our case, the smallest translation model has about 50 million parameters. Improving latency thus boils down to increasing memory bandwidth alone.

Quantization can significantly reduce inference cost. Binarization is an extreme case where both the weights and activations of a matrix multiplication (matmul) are quantized to a single bit. Compared to the Brain floating-point format (bfloat16) [1] [4], binarization reduces the weight size by $16\times$, thus significantly lowering the memory and communication overhead. Moreover, a binarized matmul can be carried out by XNOR operations followed by a population count, which is estimated to be $256\times$ more energy-efficient than the bfloat16 counterpart [39].

Prior work shows that BERT can be binarized for pretraining [5, 33, 28]; however, it is important to note that the BERT and MT models, which both use Transformer as their core [37], are very different. One key difference is the architecture: while an MT model has both an encoder and a decoder, BERT only has an encoder. This difference can impact the quality of encoder quantization because every cross attention layer in the decoder requires outputs from the encoder. Another difference is that MT model inference produces a sequence of text, while BERT performs a single text classification. This is critical because each word in the output translation sequence affects the generation of the next word. The sampling distribution of a word is therefore crucial and should be preserved after binarization, but for BERT, only the peak of the logits needs to be preserved. Due to these differences, directly applying BERT binarization techniques to MT can easily result in a lower quality model.

In this work, we investigate binarized Transformer for neural machine translation, which, to our knowledge, is the first study on this topic. Each Transformer block contains an attention layer and a feed-forward network (FFN). We binarize the weights and activations separately so we can study how each one affects the quality of the model. We found that binarizing weights did not significantly affect accuracy, but that traditional methods for binarizing activations led to poor performance due to activation magnitude explosion. Then, we propose a new method for activation binarization that uses a simple scaling factor and additional residual connections.

To understand the scaling behavior of the proposed 1-bit Transformer in practice, we further evaluate it on our in-house production-scale translation dataset that contains three billion sentence pairs. We for the first time demonstrate that the 1-bit weight Transformer scales and generalizes similarly well as the float one, even on the out-of-domain data. We also analyze sentences sampled from both models' outputs and find that the 1-bit Transformer generates a similar translation quality as its float counterpart. Binarization can therefore be a potential candidate for future MT model serving.

## 2 Related Work

The success of Transformer has spurred an active body of work to quantize it to lower precision. In this section, we review a subset of these efforts that inspired our approach.

---

[4] In the remaining paper, "float" refers to bfloat16.

**Transformer quantization.** Much of the prior effort focused on 8-bit Transformer. Bhandare et al. [8] reported a less than 0.5 BLEU drop on the WMT14 En-De translation task with 8 bits. Prato et al. [32] showed an 8-bit Transformer preserved the translation quality. For non-generative tasks, Zafrir et al. [38] quantized BERT to 8-bit with marginal quality loss. When pushed down to 4 bits, though Prato et al. [32] reported an 8 BLEU degradation for MT, Aji and Heafield [2] reported almost no BLEU loss by using a logarithmic quantization scheme.

The exploration on 1-bit Transformers centered around BERT. Usually binarization is directly applied and the focus is on improving the training recipe. Bai et al. [5] initiated the attempt by splitting a ternary BERT into a binary one, then fine-tuning. It achieved 41% average accuracy on the GLUE benchmarks. Qin et al. [33] proposed to distill each intermediate layer outputs from a floating-point model. Recently, Liu et al. [28] proposed to incrementally quantize the model, e.g., from 32-bit to 4-bit to 2-bit, finally to 1-bit, and it improved the GLUE accuracy to 73.5%.

**Binarized vision models.** Courbariaux et al. [12] pioneered the investigation on binarized deep neural nets. Recently, PokeBNN [39] established a pareto SOTA on the ImageNet recognition task. We inherit the binarization functions and training recipes from PokeBNN.

**Generalizability.** Hooker et al. [19] show that compressed models do not generalize well on out-of-domain (OOD) data. We are particularly interested in evaluating BMT under OOD settings and analyze its generalizability.

## 3 Algorithm and Model Architecture

In this section, we introduce the methodology of binarizing a Transformer-based MT model. We first define the binarization equations, then show that directly applying the equations to Transformer will produce an inferior model quality because of the dot-product variance inflation. A scaling factor is then proposed as a solution to this problem, and we discuss using LayerNorm [4] to replace fixed scaling factors. Finally, we combine and present the architectural changes that are necessary to improve the binarized model quality.

### 3.1 Binarization Equations

We follow the approach defined in PokeBNN [39] and AQT [24], which includes an important hyperparameter "$B$". The function of casting floating-point values into binary values is

$$\text{clip}\,(x, x_{min}, x_{max}) := \min\,(x_{max}, \max\,(x_{min}, x))$$
$$x_b := \left(\text{floor}\,\left(\text{clip}\,\left(\frac{x}{B}, -1 + \epsilon, 1 - \epsilon\right)\right) + 0.5\right) \times B$$

where $x$ is the input tensor, $\epsilon$ is a small floating-point number that prevents overflow when taking the floor, and $B$ is the binarization bound. In the backward propagation, the floor function is ignored, i.e., $\frac{\partial \text{floor}(x)}{\partial x} := 1$, known as the straight-through estimator [12]. The gradient of the entire binarization function is then $\frac{\partial x_b}{x} = 1_{x \in [-B, B]}$, otherwise zero. The bound $B$ therefore serves as a hyperparameter that controls the range of the input values that will have non-zero gradients. Note that $B$ also serves as a scaling factor for the outputs since the binarization function maps $x \rightarrow \left\{-\frac{B}{2}, +\frac{B}{2}\right\}$. The bound $B$ can also generalize to a vector, depending on the granularity of binarization. The finest granularity, however, is one bound value for each dot product, i.e., per contraction dimension, so that the binarized matrix multiplication can be accelerated.

For a dense layer in Transformer of the form $A \cdot W$, where $A^{N \times d_{\text{model}}}$ is the input activations and $W^{d_{\text{model}} \times d_k}$ is the model weights, we instead compute a binarized matmul $A_b \cdot W_b$. Throughout the experiments we apply binarization bound $B_W$ and $B_A$ for weights and activations, respectively.

$$B_W = \max(\text{abs}(W), \text{axis} = d_{\text{model}}), B_A = \max(\text{abs}(A), \text{axis} = d_{\text{model}})$$

where `axis` is the dimension along which `max` is taken. Using one axis means the bound is per channel and per example [24]. Both $B_A^{N \times 1}$ and $B_W^{1 \times d_k}$ are vectors that contain maximum absolute values along the contraction dimension. Note that the weight binarization bound $B_W$ is static in inference though it is updated in every training iteration. The activation bound $B_A$ is dynamic.

## 3.2 Variance Inflation in Binarization

We start by applying the binarization function to feed-forward networks (FFNs), leaving other modules as float. We observe that directly binarizing the weights preserves the model quality, but binarizing the input activations causes the training to *not* converge in the context of machine translation. To understand the reason of this behavior, we analyze the variance of the dot product magnitude with and without binarization. Our analysis reveals that binarizing both weights and activations will statistically inflate the magnitude, leading to abnormal signal propagation within the neural network [10]. We present the details of this analysis as follows.

Let each weight of a dense layer be randomly initialized and sampled from a zero-mean normal distribution, $w \sim \mathcal{N}(0, \sigma_w^2)$. Assume each input activation is independent of the weights and identically distributed as $a \sim \mathcal{N}(0, \sigma_a^2)$. After applying the binarization function, both $w_b$ and $a_b$ are still centered at zero and have an equal probability of being either $-\frac{B}{2}$ or $+\frac{B}{2}$, namely, they follow the probability mass function defined as $\Pr(x_b) = \begin{cases} \frac{1}{2} & x_b = -\frac{B}{2} \\ \frac{1}{2} & x_b = +\frac{B}{2} \end{cases}$. Hence the variance of a binarized multiplication is

$$\mathrm{Var}\left(a_b \cdot w_b\right) = \mathbb{E}\left[a_b^2\right] \cdot \mathbb{E}\left[w_b^2\right] - \mathbb{E}^2\left[a_b\right] \cdot \mathbb{E}^2\left[w_b\right] = \sum_{a_b} a_b^2 \cdot \Pr\left(a_b\right) \cdot \sum_{w_b} w_b^2 \cdot \Pr\left(w_b\right) - 0 = \frac{B^4}{16}$$

The variance of a binarized dot product is then $\mathrm{Var}\left(A_b \cdot W_b\right) = \sum_{n=0}^{D-1} \mathrm{Var}_n\left(a_b \cdot w_b\right) = \frac{B^4}{16} \cdot D$, where $D$ is the dimensionality of the dot product, i.e., the hidden projection dimension in an FFN, and $n$ is the index of each entry in the vector.

Following the same analysis, the variance of a floating-point dot-product is $\mathrm{Var}\left(A \cdot W\right) = \sigma_a^2 \cdot \sigma_w^2 \cdot D$. Note that the commonly used Xavier initializer [17] equalizes the variance of the activations across layers. $\sigma_w^2$ will therefore be initialized as $\frac{1}{D}$, so $\mathrm{Var}\left(A \cdot W\right) = \sigma_a^2$, which is usually at the scale of 1.

Meanwhile, the common binarization bound is $B \in [1, 3]$ [12, 39, 7]. Our Transformer FFN employs a hidden projection dimension $D = 4096$ throughout the experiments. Therefore, $\mathrm{Var}\left(A_b \cdot W_b\right) \gg \mathrm{Var}\left(A \cdot W\right)$. Binarization heavily inflates the dot product variance by at least $256\times$, which will be reflected in the magnitude of the dense layer outputs. Also note that $\mathrm{Var}\left(A_b \cdot W_b\right) \propto D$, indicating that Transformer with a larger width will potentially suffer more from the convergence issue.

## 3.3 A Scaling Factor as the Solution

Inspired by the scaling factor $\sqrt{d_k}$ in the scaled dot-product attention $\mathrm{Attention}\left(Q, K, V\right) = \mathrm{softmax}\left(\frac{QK^T}{\sqrt{d_k}}\right)V$ in the original Transformer [37], we propose a scaling factor for each binarized dense layer, i.e.,

$$\mathrm{Dense}\left(A_b\right) = \frac{A_b \cdot W_b}{s}$$

The scaling factor $s$ is a hyperparameter that suppresses dot-product variance inflation, while in the attention layer $\sqrt{d_k}$ prevents the dot products from entering small-gradient regions of the softmax function. According to the analysis in Section 3.2, its value is estimated to be $s \propto \sqrt{D}$ in order to cancel the multiplicative effect from $D$ on the variance.

To verify how the magnitude of the scaling factor affects the training loss, we sweep $s$ in Section 5. In practice, $s \geq 64$ can make the training converge.

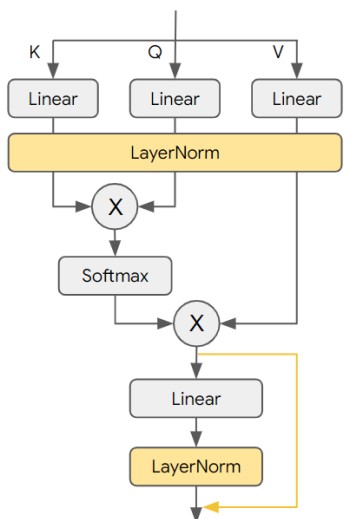

Figure 1: BMT Multi-Head Attention — Differences from the original Transformer are highlighted (in yellow). All linear projections and einsums can be binarized.

## 3.4 Replacement of Scaling Factor with LayerNorm

While the scaling factor $s$ enables the binarization of FFNs, it requires hyperparameter tuning, which can be challenging for billion-parameter translation models. To address this deficiency, we propose using layer normalization

(LayerNorm) [4] as a drop-in replacement for the scaling factor, which has the form of $\text{LN}(x) = \frac{x - \mathbb{E}[x]}{\sqrt{\text{Var}(x)+\epsilon}} \cdot \gamma + \beta$, where $\gamma$ and $\beta$ are learnable parameters. Besides the fact that $\gamma$ can incorporate the scaling factor $s$, LayerNorm also has the following advantages.

*The scaling factor is now dynamic and adaptive during training.* The binarization function employs a dynamic bound $B$, so $\text{Var}(A_b \cdot W_b)$ varies. The learnable parameter $\gamma$ in LayerNorm can better capture the changes in the dot product variance and hence properly normalize it.

*LayerNorm also redistributes the input activations.* It enables the binarization of a tensor with all positive values. A directly binarized FFN is $\text{FFN}(A) = \max(0, A_b W_{1b} + b_1)_b W_{2b} + b_2$, where $W_1, b_1$ and $W_2, b_2$ are the weights and biases for the first and second dense layer, respectively. One may note that the activations $\max(0, A_b W_{1b} + b_1)$ are all positive. The binarization function will then map the entire tensor to a constant $+\frac{B}{2}$, which undermines the model training. With the help LayerNorm, however, the activations are redistributed and more balanced in terms of the number of positive and negative values. This enables the normal $\{-1, +1\}$ (bipolar) binarization of the second dense layer. Qin et al. [33], Liu et al. [28] used $\{0, 1\}$ binarization instead in binarized BERT to overcome the issue of constant positive values. It yields a ternary matrix multiplication since $A \in \{0, 1\}^{N \times D}$ and $W \in \{-1, +1\}^{D \times K}$, which incurs nontrivial additional overhead if computed on binary hardware accelerator. The complete proposed 1-bit FFN has the structure of

$$\text{FFN}(A) = \text{LN}\left(\text{LN}\left(\max(0, A_b W_{1b} + b_1)\right)_b \cdot W_{2b} + b_2\right)$$

When proceeding to the attention binarization, we add a LayerNorm to the output of each linear projection layer for the same reasons. We verified in Section 5 that a dynamic and adaptive scaling factor in LayerNorm indeed outperformed a fixed one.

## 3.5 Residual Connection in Attention Layers

In attention layers, we also add a shortcut connection to the output linear projection layer. Combined with the additional LayerNorm, the output projection then becomes $\text{Out}(A) = \text{LN}(A \cdot W) + A$. In BNNs, gradients of a binarized layer are approximated due to the straight-through estimator. This will eventually lead the optimization into a different direction as we stack more binarized layers. Liu et al. [26] proposed adding additional residual connections in BNNs, which became a useful method for partially addressing this issue. We therefore adopt it in our model. Note that this modification is unnecessary for QKV (query, key, value) linear projections. The shortcut around the entire attention layer in the original Transform serves the same purpose. We will also demonstrate the effectiveness of the shortcut connection in the ablation study in Section 5.

The complete modified attention architecture is shown in Figure 1, where we highlight the differences from the original one. The extra layer normalization and shortcut connection are both elementwise. Their overhead is small, especially comparing to the benefits of binarization.

## 4 Experiments

In this section, we empirically evaluate our proposed binarized Transformer on MT tasks at difference scales. To investigate the impact of binarizing different layers, we first train a standard 6-layer encoder-decoder (6L6L) Transformer on the WMT2017 De-En translation dataset [9] and evaluate it on the WMT2014 De-En dataset. We then choose the 1-bit weight model variant and study its practical scaling law on in-house translation datasets. We also compare the translation qualities of both 1-bit and float models. Throughout the experiments, the embedding table and the prediction head layer are not binarized.

### 4.1 WMT Results

We binarize five different matmuls in a Transformer. In an attention layer there are (1) QKV linear projections; (2) activation-activation matmul between queries and keys (QK Einsum); (3) activation-activation matmul between attention scores and values (Score-V Einsum); (4) output linear projection. In an FFN there are two dense layers of the same type. To study their individual impact, we binarize their weights and activations separately. In our experiments we use the following training details.

**Model.** We use a 6L6L Transformer as the base model. Embedding dimension is 1024. Each multi-head attention layer has 16 heads, with a dimension of 1024 for QKV if combining all the heads. The hidden projection dimension in FFNs is 4096. Dropout layers has a dropout rate of 0.1.

**Scheduler.** We adopt a three-stage training scheme, where the learning rate (LR) of each stage decreases from base to zero following a cosine decay. A quantization event starts at the beginning of each stage. We first train the model in float. In the second stage, all weights will be binarized. In the last stage, both weights and activations will be binarized.

Table 1: BMT results on the WMT dataset. Training uses WMT2017 De-En and evaluation uses WMT2014 De-En. Binarized activations or weights are labeled by checkmarks. Unlabeled tensors remains bfloat16. 1-bit weights models have 25MB of weight storage size while the float one has 399MB, $\sim 16\times$ compression. As a comparison, the baseline model (last row) directly applies XNOR-Net style [34] binarization used in previous works [33, 28], sign function followed by a normalization. BLEU evaluation employs a beam size of 4.

| | ATTENTION 1-BIT | | | | | | FFN 1-BIT | | METRICS | |
|---|---|---|---|---|---|---|---|---|---|---|
| | $A_{\text{QKV}}$ | $W_{\text{QKV}}$ | $A_{\text{OUT}}$ | $W_{\text{OUT}}$ | QK | SCORE-V | $A_{\text{FFN}}$ | $W_{\text{FFN}}$ | VAL LOSS | BLEU |
| FLOAT | | | | | | | | | 1.39 | 26.35 |
| BMT-1 | | ✓ | | ✓ | | | | ✓ | 1.38 | 25.93 |
| BMT-2 | | | | | | | ✓ | ✓ | 1.40 | 25.44 |
| BMT-3 | | ✓ | | ✓ | | | ✓ | ✓ | 1.51 | 24.11 |
| BMT-4 | ✓ | ✓ | | ✓ | | | ✓ | ✓ | 1.72 | 21.55 |
| BMT-5 | | ✓ | ✓ | ✓ | | | ✓ | ✓ | 1.60 | 21.06 |
| BMT-6 | ✓ | ✓ | ✓ | ✓ | | | ✓ | ✓ | 1.89 | 17.87 |
| BMT-7 | | ✓ | | ✓ | ✓ | | | ✓ | 1.76 | 18.27 |
| BMT-8 | | ✓ | | ✓ | ✓ | ✓ | | ✓ | 2.81 | 9.42 |
| BASE [33, 28] | | | | | | | ✓ | ✓ | 8.07 | 0.21 |

**Training.** We apply knowledge distillation (KD) during training. KD replaces the ground truth label in the cross-entropy loss function with the softmaxed logits from the teacher model, so it is optional for users. Adam optimizer [22] is used with $\beta_1 = 0.9$ and $\beta_2 = 0.98$. No weight decay is applied. Batch size is 1024. Base learning rate is 0.001. The first LR cycle has 50000 steps, others have 88339 steps. We train the model with a $4 \times 8$ TPU topology.

**Observations.** The evaluation results on WMT2014 De-En translation dataset is shown in Table 1. We mainly rely on the validation loss for comparing the model quality since BLEU score has a higher variation [16]. From the table we have the following key observations.

**Weight-only binarization preserves the model loss.** The float 6L6L Transformer baseline has a 1.39 validation loss. In contrast, binarizing all dense layer weights (in both attention layers and FFNs) produces an even lower loss (1.38, BMT-1), though the BLEU score slightly drops by about 0.4. Both metrics indicate that the 1-bit weight model has a similar translation quality to the float baseline. Binarization has the potential to compress the model storage size by $16\times$ while preserving the quality.

**FFN binarization produces promising results.** Binarizing the entire FFN, i.e., both activations and weights, while leaving other layers float, again yields a similar validation loss (1.4, BMT-2) compared with the float baseline. With our proposed BMT, it is the first time on machine translation tasks that binarizing FFN activations can preserve the loss. This intriguing 1-bit FFN variant can be potentially useful for large language models. Combing with 1-bit all dense layer weights further downgrades the loss to 1.51 (BMT-3) and a 2.2 lower BLEU score in contrast to the float model. Overall, FFN binarization demonstrates a promising potential.

**Attention activations are the key bottleneck to high binary model quality.** On top of the 1-bit weights and 1-bit FFN activation model variant, further binarizing input activations in all dense layers in the attention layer (BMT-6; this includes keys, queries, values and input activations to the output projection dense layer) leads to a 1.89 loss. This is by far the largest drop in model quality. Binarizing each individual activation tensor therein leads to at least 0.3 degradation in loss (BMT-4 and 5). In addition, binarizing the two activation-activation matmuls (query-key einsum operation and

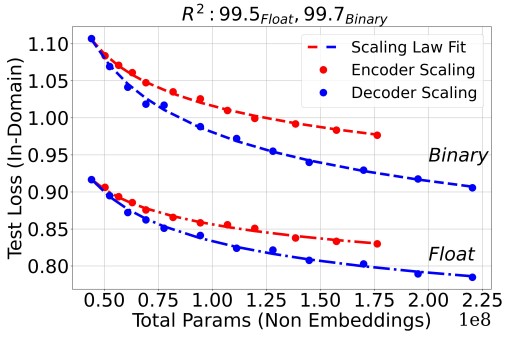
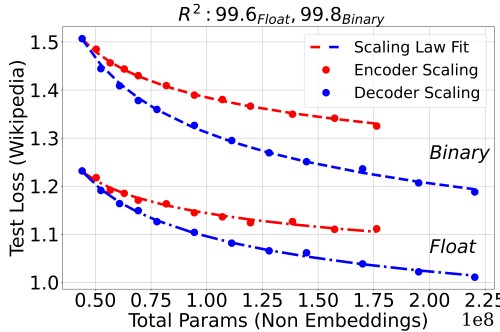

(a) $\{p_e, p_d\} : \{0.18, 0.31\}_{\text{Float}}, \{0.16, 0.28\}_{\text{Binary}}$. In-domain evaluation.

(b) $\{p_e, p_d\} : \{0.13, 0.25\}_{\text{Float}}, \{0.13, 0.25\}_{\text{Binary}}$. Out-of-domain evaluation.

Figure 2: Scaling law study on both in-domain and out-of-domain data — On in-domain data, scaling law fits achieve $R^2$ values of 99.5 and 99.7 on float and binary models respectively. On out-of-domain data (Wikipedia), $R^2$ values are 99.6 and 99.8 respectively. Scaling law fit on all the evaluation datasets, along with slopes ($p_e$ and $p_d$) is presented in Figure 7 and Figure 8 (Appendix A).

attention score-value einsum operation) are particularly challenging. The 1-bit weights model with both activation-activation matmuls binarized additionally produces only $9.4$ BLEU score (BMT-8). Attention layer activations are the current bottleneck to a fully binarized translation model.

## 4.2 Scaling Law Study

Though Section 4.1 shows promising results, an unanswered question is whether the performance degrades when binarized Transformers are scaled up. Neural language model loss is known to follow a power law as its model size scales up [21], known as the "scaling law". It is widely adopted for predicting the performance of models at scale. Prior work shows 8-bit and 4-bit language models are subject to a certain scaling law, but this has not yet been established for models with 3-bits or lower [13]. We therefore conduct a scaling law study on both float and the proposed binarized models on our in-house translation dataset and compare their difference. Similar to Ghorbani et al. [16], we train a set of translation models and fit the losses using the following equation: $L(N_e, N_d) = \alpha \left( \frac{\bar{N}_e}{N_e} \right)^{p_e} \left( \frac{\bar{N}_d}{N_d} \right)^{p_d} + L_\infty$, where $L$ is the per token loss, $N_e$, $N_d$ are the number of encoder and decoder parameters respectively. $L_\infty$ is the irreducible loss that the model attains if it has infinite capacity. $\bar{N}_e$ ($\bar{N}_d$) is the number of parameters in the baseline 6L6L Transformer, which act as normalization constants for numerical stability in the curve fitting process. For tractability purposes, we examine scaling laws for only weight-binarized models. Weight-only model compression can also be leveraged for linear improvements in latency [36] and $16\times$ improvements in memory consumption (compared to blfoat16).

**Dataset.** To investigate the scaling behavior of the binary models in a capacity limited regime, i.e., performance is not bound by training data, we use our large in-house parallel corpora for English to German (En $\rightarrow$ De) direction. The training set contains 3 billion web-crawled sentence pairs. We are also particularly interested in evaluating BMT with the out-of-domain (OOD) setting and assessing its generalizability, as previous research in the image domain demonstrated that compressed models (weight pruned or quantized) have a much larger quality drop on OOD data than their uncompressed counterparts, i.e., model compression amplifies brittleness [19]. As such, to have a robust evaluation of BMT, we use eleven evaluation sets, one of which is in-domain (ID) and is similarly distributed as the training set, and the rest are OOD. For ID, we sample 2000 training examples and remove them from the training data. The ten OOD evaluation sets are divided into four categories (i) Web Domain (ii) News Domain (iii) Wikipedia (iv) Patents. Furthermore, they are either "source-original" or "target-original". The source-original datasets have a natural source side (English) while the target side (German) is human or machine translated. The target-original datasets have the natural target side (German), then back translated into source English sentences. We do this differentiation to investigate the impact of binarization on "style" of sentences since natural language exhibits rich

diversity as opposed to simple and literal (*translationese*) sentences [14] (More details are provided in Appendex A.1).

**Models & Training.** We train two sets of Transformers, namely, encoder-scaling and decoder-scaling models. The encoder-scaling models have a fixed depth of 6 layers in the decoder while scaling up the encoder depth in sizes of $\{6, 8, 10, 12, 14, 18, 22, 26, 30, 36, 42, 48\}$ layers, for a total of 12 models. Same for the decoder-scaling ones, whereby the decoder depth is scaled up in similar ways. Due to the sufficiency in training data, we did not use label smoothing during training. The binary models are trained without KD. (Appendix A.2 has more details on hyper-parameters and training).

**Observations.** Figure 2 compares the scaling curves of the binary and float models on both ID and OOD datasets, more in Appendix A.3. Figure 3 compares their training vs. In-domain test loss. We make the following observations:

**Binary models demonstrated similar scaling behaviors as their float counterpart for both encoder and decoder scaling.** The exponent of the fitted power law for binary models in Figure 2a ($p_e = 0.16$, $p_d = 0.28$) is only slightly below float ones ($p_e = 0.18$, $p_d = 0.31$), indicating the binary model loss improves fast as the parameter count increases. This trend also holds for OOD Wikipedia dataset in Figure 2b. Binary models generalize just as well on OOD data as float models (scaling law fits on all the OOD evaluation datasets is in Appendix A.3). We also note a gap between binary and float model losses, a phenomenon not observed from WMT experiments. We hypothesize that this is because the in-house production-scale datasets are more challenging.

**For the same training loss, binary and float models achieve the same generalization performance.** As shown in Figure 3, binary and float model losses align well on a straight line, and almost overlap in the $0.95 \sim 1.0$ region. There are no measurable differences detected in the inductive biases of the two model classes. Also, binary models require fewer parameter bits to achieve a certain performance level. For example, a 6L42L binary Transformer with 195M parameters (195M bits) has a $4.3\times$ smaller size than a 6L8L float one with 52M parameters (832M bits) while having the same loss. Such memory savings are especially advantageous when the models are deployed in a resource-constrained environments [36].

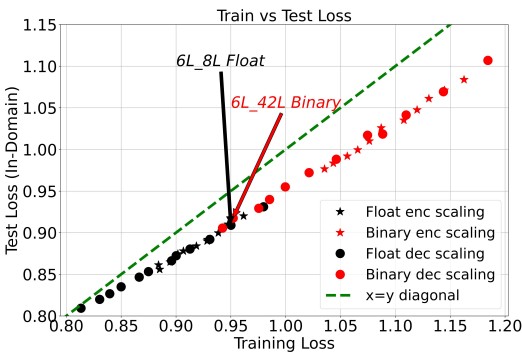

Figure 3: Training vs. ID Test loss. We observe similar linear relationship between training and test losses of all evaluation datasets.

### 4.3 Generation Quality

We examine the MT model generation quality in Figure 4 using two decoding strategies: a) Beam Search Decoding; b) Minimum Bayes Risk (MBR) decoding [23].

**Beam search.** Sample quality from Beam search decoding is evaluated with standard de-tokenized BLEU scores [30] using sacreBLEU library [31][5].

**MBR.** Freitag et al. [15] show that beam search decoding selects samples with high probability rather than high quality, especially for large models, as measured by human evaluations. They propose MBR-based decoding strategy defined as $h^{\text{MBR}} = \arg\max_{h \in \mathcal{H}} \frac{1}{|\mathcal{H}_{\text{model}}|} \sum_{y \in \mathcal{H}_{\text{model}}} u(h, y)$, where $h^{\text{MBR}}$ is the decoding from the model given source sentence $x$, $\mathcal{H}_{\text{model}}$ is the set of hypotheses sampled from the model $p(.|x)$ and $u$ is a utility function that evaluates quality of a hypothesis $h$ against reference $y$. Freitag et al. [15] demonstrate effectiveness of the BLEURT model [35] for the utility function. BLEURT is a regression model that relies on the concatenation of hypothesis $h$ and reference $y$ and generates a scalar score between [0,1], measuring the hypothesis quality irrespective of the sentence structure, length or word overlap with the reference. In the same way, we use MBR decoding with BLEURT as the utility function to decode a sequence given the source sentence. To measure the sample quality, BLEURT($h, r$) is calculated between the decoded hypothesis ($h_{\text{MBR}}$) and the reference ($r$) for a given source sentence ($x$), then averaged across the evaluation set.

---

[5]Beam size=4, length penalty=0.6. case.mixed + numrefs.1 + smooth.exp + tok.13a

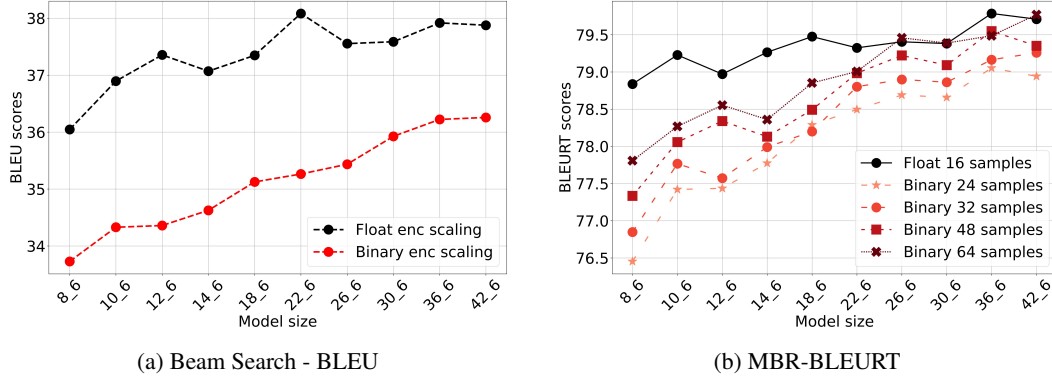

|(a) Beam Search - BLEU|(b) MBR-BLEURT|

Figure 4: Comparison on translation qualities between binarized and float models for encoder-scaling. (a) Beam Search Decoding: BLEU scores on In-Domain Test set (b) MBR Decoding: BLEURT scores on In-Domain Test set

**Observations.** Figure 4a shows BLEU scores of encoder-scaling models (i.e., decoder depth=6, varying encoder depth). Figure 4b plots BLEURT scores for encoder-scaling models, where the baseline is float models using MBR decoding with 16 samples. We observe the following:

**Binary models can achieve the same BLEU score as float models with a smaller size.** Figure 4a shows that the BLEU score of binary models will consistently improve as the model size increases. Although binary models are 2-3 BLEU worse than float ones at the same model depth, the 30L6L binary model achieves the same BLEU as the 8L6L float model, while being $6.7\times$ smaller in size.

**Increasing the sample size can match the generation quality of binary models with float models.** In Figure 4b, a larger sample size consistently produces a higher generation quality for the binary models. At $4\times$ the sample size, i.e., 64 samples, the binary model quality approximately matches float models. Besides, the BLEURT score of binary models also improves as the model size increases.

## 5 Ablation Study

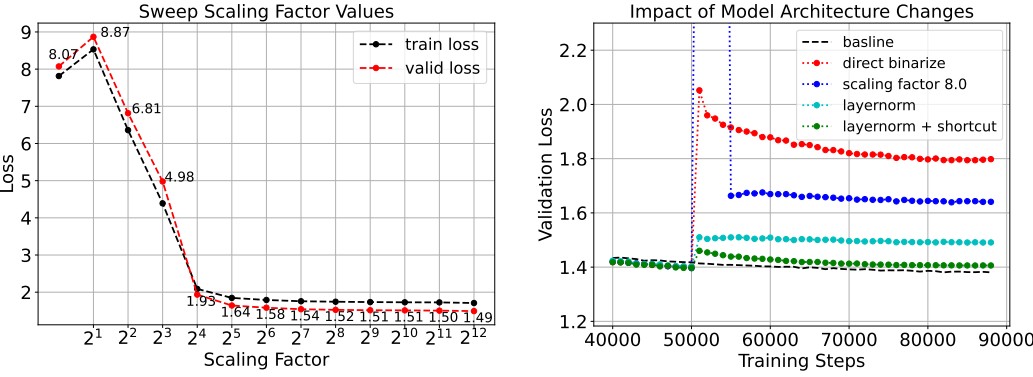

(a) Models losses with different scaling factor $s$.     (b) Losses of different attention out linear architectures.

**Scaling factor ablation.** We binarize the FFN only and sweep the scaling factor $s$ as a power of two from 1 (equivalent to no scaling factor applied) to 4096. We plot the final training and validation losses in Figure 5a. The model losses drop steeply when increasing $s$ to 64. Models with $s \leq 8$ produce almost random translation quality. Large scaling factors indeed address the convergence issue. The loss begins saturated at $s = 64$ and is only slightly worse than the float baseline (1.39). This exactly matches our expectation that $s \propto \sqrt{D}$. When $s > 64$, the model loss keeps improving slightly. We hypothesize that this is because the bound $B$ is dynamic. Even a small variation on $B$ will change the theoretical optimal $s$ by a large margin since $\mathrm{Var}\,(A_b \cdot W_b) \propto B^4$.

**BMT attention layer ablation.** We only binarize the attention output projection linear layer. We train the model for $88339$ steps, with binarization events started at step $50000$. We plot the loss curves from step $40000$ in Figure 5b. Applying a fixed scaling factor achieves an almost $0.2$ loss improvement. This is consistent with previous observations where a scaling factor helps with convergence. The LayerNorm, as a drop-in replacement for the scaling factor, not only makes the model converge to a better loss, but also recovers the loss much faster after binarization. This is expected because $\gamma$ in the LayerNorm is learnable and can better adapt to the dynamic bound $B$ as analyzed in Section 3.4. The loss almost saturates after binarization. Adding a shortcut around the output projection removes the information bottleneck. It helps the model converge to approximately the same quality as the float baseline.

## 6 Conclusion

The proposed method enables binarization for machine translation. The simple yet effective scaling factor is the key. Binary Transformers have a similar scaling behavior of translation quality as float models. Binarization can thus be a potential candidate for future model serving.

Unanswered questions: How to better binarize attention einsums? Which is better for scaling up a binary Transformer, depth or width? What will be a good mixed-precision scheme?

## Acknowledgments and Disclosure of Funding

This work is supported in part by NSF Award #2007832.

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

# A   Scaling Law Study Details

## A.1   Dataset

A concise view of evaluation datasets used for scaling laws (Section 4.2) is shown in Table. The ten OOD evaluation datsets span four categories (i) Web Domain (ii) News Domain (iii) Wikipedia (iv) Patents. They are either "source-original" or "target-original". There are two source-original and one target-original dataset in Web Domain, one source-original each in Wikipedia and Patents domain. We use publicly available WMT newstest2019 [6] and WMT newstest2021 [3] for News Domain. Within this domain, we have five datasets: source-original, target-original, source-original-paraphrased [14] and source-original-high-quality [14] from WMT newstest2019 [6], and wmt-reference-C from WMT newstest2021 [3].

| Dataset Name | Domain | Type | Source |
| --- | --- | --- | --- |
| Train Subset | Web | mixed | In-house |
| Patents | Patents | mixed | In-house |
| Web domain 1 | Web | source-original | In-house |
| Web domain 2 | Web | source-original | In-house |
| Web domain 3 | Web | target-original | In-house |
| Wikipedia | Wikipedia | source-original | In-house |
| wmt-high-quality | News | source-original | WMT newstest2019 [14] |
| wmt-refC | News | source-original | WMT newstest2021 Ref-C [3] |
| wmt-paraphrased | News | source-original | WMT newstest2019 [14] |
| wmt-src-orig | News | source-original | WMT newstest2019 [6] |
| wmt-tgt-orig | News | target-original | WMT newstest2019 [6] |

Table 2: Evaluation datasets used in Section 4.2.

## A.2   Model & Training Details

All the models in Section 4.2 have an embedding dimension of 512, a hidden projection dimension of 2048, and 8 attention heads. The embedding parameters are shared on the source and the target side. The same embedding matrix (transposed) is also used for the linear readout (softmax) parameters on the decoder side. All models are trained with Adam optimizer [22] and use cosine learning rate schedule. Due to the sufficiency in training data, we did not use label smoothing during training. In our experiments, enabling label smoothing resulted in poor development set performance across all the models. Training and Learning rate profiles of one model (6 encoder, 8 decoder layers) are shown in Figure 6. Float models are trained for 5 epochs, and binary models are trained for 9 epochs in two stages: float stage and a binarization stage. An independent but identical learning rate schedules are used (with warmup) in both the stages of the binary model training. We note that a significant amount of training (i.e. loss reduction) for binary models happens in the final 10 steps when the learning rate is extremely small. Raw values of last 15 steps of learning rates are [5.0e-7, 3.1e-7, 1.6e-7, 6.4e-7, 1.0e-8, {2.5e-15}x10]. We also tune binary models with a constant learning rate of values in {1e-8, 1e-11, 1e-15} for the last epoch (overriding the original schedule), however we observe degradation in the quality (loss plateaus). This phenomenon of significant learning in the final stages of binary models' training at extremely small learning rates is also observed by Liu et al. [27], Zhang et al. [39]. We leave further investigation of this behavior to future work.

## A.3   Scaling Law Fit

Scaling law fit on all ten OOD evaluation datasets is shown in Figure 7. The slopes $p_e$ and $p_d$ are shown in Figure 8 and Table 3.

# B   Generation Quality

Generation quality for decoder-scaling models is shown in Figure 9. We observe similar behavior as seen for encoder-scaling models in Section 4.3. BLEU scores for binary models are 2-3 BLEU points

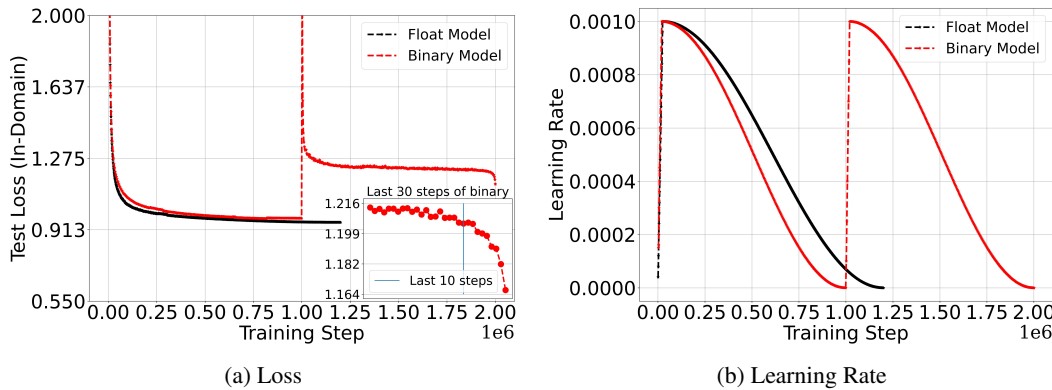

|  | (a) Loss | (b) Learning Rate |

Figure 6: Test loss and learning rate profiles of a 6L8L float and binary model as the training progresses.

| Dataset | Float models | | Binary models | |
|---|---|---|---|---|
| | $p_e$ | $p_d$ | $p_e$ | $p_d$ |
| Train Subset | 0.18 | 0.31 | 0.16 | 0.28 |
| Patents | 0.20 | 0.30 | 0.19 | 0.32 |
| Web Domain 1 | 0.14 | 0.25 | 0.14 | 0.27 |
| Web Domain 2 | 0.19 | 0.37 | 0.16 | 0.30 |
| Web Domain 3 | 0.12 | 0.18 | 0.14 | 0.23 |
| Wikipedia | 0.13 | 0.25 | 0.12 | 0.25 |
| wmt-high-quality | 0.20 | 0.31 | 0.18 | 0.30 |
| wmt-refC | 0.24 | 0.34 | 0.17 | 0.27 |
| wmt-paraphrased | 0.14 | 0.36 | 0.12 | 0.31 |
| wmt-src-orig | 0.22 | 0.37 | 0.23 | 0.36 |
| wmt-tgt-orig | 0.15 | 0.22 | 0.12 | 0.20 |

Table 3: Tabular representation of the same data ($p_e$ & $p_d$) as shown in Figure 8.

worse than the respective float models at the same model depth. MBR-BLEURT based decoding quality increases consistently by increasing the sample size.

### B.1 Translation Samples

We generate several En-De translation samples as follows from both float and binary 6L10L encoder-decoder models. Inputs are taken from the WMT2017 dataset.

**Example 1**

- **Source:** The notice for the Nottingham East Labour meeting on Friday stated that "we want the meetings to be inclusive and productive."

- **Reference:** In der Mitteilung für das East Labour in Nottingham Treffen am Freitag heißt es: "Wir wollen, dass die Treffen integrativ und produktiv sind".

- **Float output:** In der Mitteilung für das Nottingham East Labour-Treffen am Freitag heißt es: "Wir wollen, dass die Treffen inklusive und produktiv sind."

- **Binary output:** Die Bekanntmachung für das Nottingham East Labour-Treffen am Freitag erklärte: "Wir wollen, dass die Sitzungen inklusive und produktiv sind."

**Example 2**

- **Source:** The Government of Wales Act 2017 gave the Welsh assembly the power to change its name.

- **Reference:** Mit dem Government of Wales Act 2017 erhielt das walisische Parlament die Möglichkeit, seinen Namen zu ändern.
- **Float output:** Der Government of Wales Act 2017 gab der walisischen Versammlung die Befugnis, ihren Namen zu ändern.
- **Binary output:** Das Government of Wales Act 2017 gab der walisischen Versammlung die Befugnis, ihren Namen zu ändern.

**Example 3**

- **Source:** Residents were seen returning to their destroyed homes, picking through water-logged belongings, trying to salvage anything they could find.
- **Reference:** Anwohner wurden dabei beobachtet, wie sie in ihre zerstörten Häuser zurück-kehrten, völlig durchnässte persönliche Gegenstände mitnahmen und versuchten zu retten, was zu retten ist.
- **Float output:** Die Bewohner wurden gesehen, wie sie in ihre zerstörten Häuser zurück-kehrten, durch verstopfte Habseligkeiten pflückten und versuchten, alles zu retten, was sie finden konnten.
- **Binary output:** Die Bewohner wurden gesehen, wie sie in ihre zerstörten Häuser zurück-kehrten, indem sie mit Wasser gefüllte Gegenstände pflückten und versuchten, alles zu retten, was sie finden konnten.

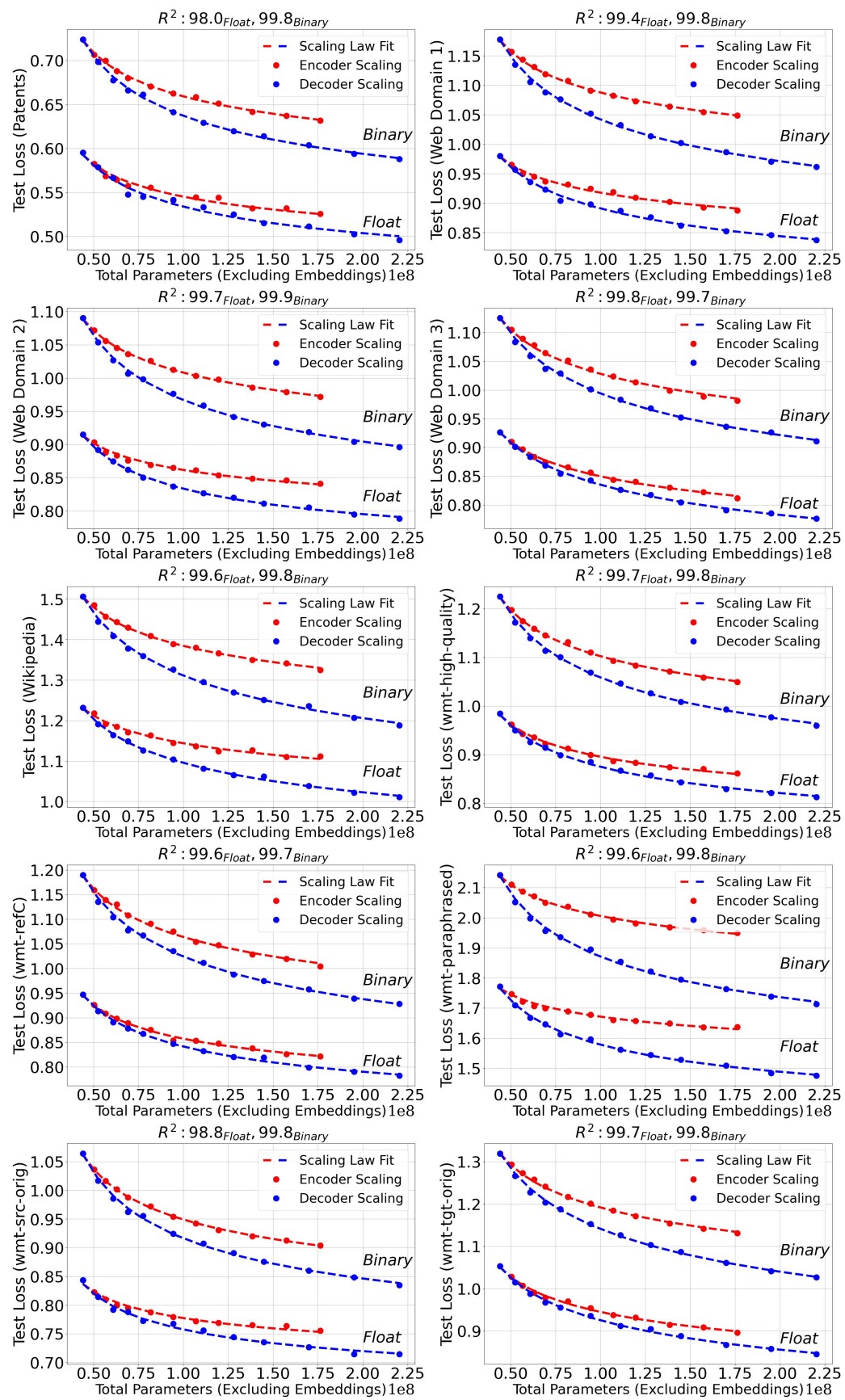

Figure 7: Scaling law studies on evaluation datasets defined in Section 4.2

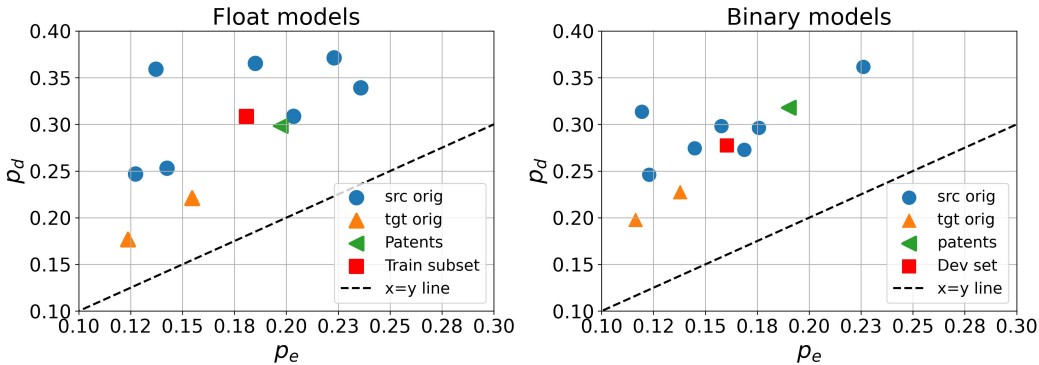

Figure 8: Encoder and Decoder scaling slopes (i.e. $p_e$ & $p_d$) as per the scaling law defined in Section 4.2. Raw values are shown in Table 3.

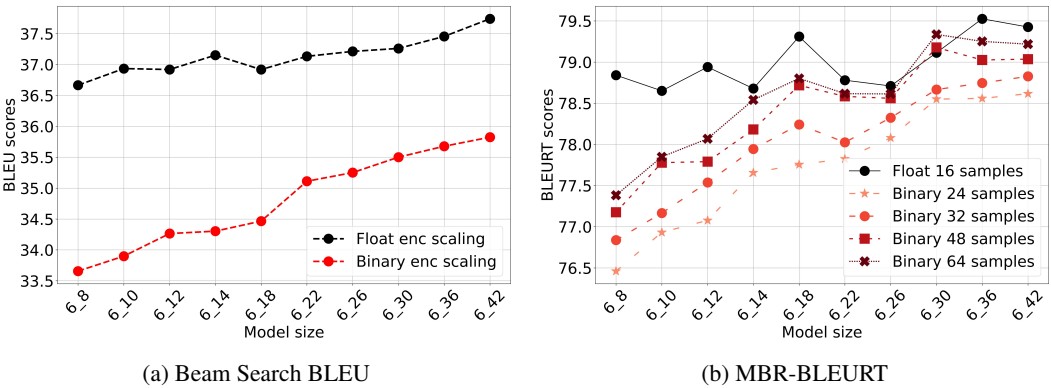

(a) Beam Search BLEU

(b) MBR-BLEURT

Figure 9: Comparison on translation qualities between binarized and bfloat16 models for decoder-scaling.

