# OpenReview forum: "Binarized Neural Machine Translation"
_NeurIPS.cc/2023/Conference — NeurIPS 2023 poster_

### Official Review · Reviewer_BTaK · 2023-06-24

**Soundness:** 4 excellent
**Presentation:** 4 excellent
**Contribution:** 3 good
**Rating:** 8
**Confidence:** 3

**Summary:**

They propose a novel binarization technique for Trans3 formers applied to machine translation (BMT), the first of its kind. They identify and
address the problem of inflated dot-product variance when using one-bit weights and activations. Specifically, BMT leverages additional LayerNorms and residual connections to improve binarization quality. Experiments on the WMT dataset show that a one-bit weight-only Transformer can achieve the same quality as a float one, while being 16× smaller in size. They further conduct a scaling law study using production-scale translation datasets, which shows that one-bit weight Transformers scale and generalize well in both in-domain and
out-of-domain settings.

**Strengths:**

1 A novel binarized NMT model is proposed, which may be useful for the production server.
2 The proposed scaling factor to mitigate the activation variance is simple and effective. And the whole model architecture looks convincing.
3 The experimental results are very solid.

**Weaknesses:**

1 How the training and inference efficiency change compared to the float model is not discussed.
2 The proposed code should include the readme, which can help the reviewer to run and check the main results.
2 typo error, line 348

**Questions:**

See above

**Limitations:**

More discussion about  the limitations should be presented.

---

> ### Author Rebuttal · Authors · 2023-08-09
>
> **Q1: How the training and inference efficiency change compared to the float model is not discussed.**
>
> Thanks for commenting on the efficiency side. It is not accessible because we are not aware of an ecosystem (accelerator combined with software stack) that supports such measurement for 1-bit models. However, there is convincing evidence that 1-bit matmuls will have high performance. For example, NVIDIA’s A100 architecture [1] shows 1-bit matmul is 8x faster than 8-bit measured by TFLOPS throughput, though requiring NVIDIA’s own assembly. Also [2] shows binary matmul is 9x-12x faster compared to 8-bit measured by latency on ARM CPUs. In that light, among other goals, our work is aiming to contribute to the assessment of whether binary ML hardware+software is a viable future direction for the ML accelerator industry.
>
> [1] NVIDIA Ampere Architecture Whitepaper. Table 3.
>
> [2] Tom B., et al., Larq Compute Engine: Design, Benchmark, and Deploy State-of-the-Art Binarized Neural Networks, MLSys’21.
>
> **Q2: The proposed code should include the readme, which can help the reviewer to run and check the main results**
>
> We really appreciate the reviewer checking out our software design, and we will write a good readme indeed. As promised in the abstract (line 12), we will open-source the code together with a detailed reproduction instruction upon acceptance.
>
> **Q3: typo error, line 348**
>
> Thanks for pointing it out. We will remove the redundant word “the”.
>
> **Q4: More discussion about the limitations should be presented.**
>
> We will make the limitations more clear. (1) As stated in line 237, the activation-activation matmul binarization quality is still not ideal. We list 8 BMT variants in Table 1, among which BMT-8 (the one with activation-activation matmul binarization) has the largest loss and BLEU drop. We imagine a lot of the future effort will be dedicated to it. (2) As stated in the last part of the conclusion, there are several unanswered questions in this work, for example, shall we scale up the depth or width for the binarized Transformer and how should we design a mixed-precision scheme using binarization and potentially other formats? We will expand these discussions in the next revision.

---

> > ### Comment · Reviewer_BTaK · 2023-08-14
> > **Rebuttar Readed**
> >
> > Thanks for your answering.

---

### Official Review · Reviewer_B1hS · 2023-07-03

**Soundness:** 3 good
**Presentation:** 4 excellent
**Contribution:** 3 good
**Rating:** 6
**Confidence:** 4

**Summary:**

The paper proposes a novel quantization scheme to binarize transformer machine translation models.  The method consists of inserting additional layer normalization for activations and also additional residual connections.  The authors demonstrate good results on the WMT test set especially for weight-only binarization, and promising results for both weight and activation binarization.

**Strengths:**

- The paper is well written and the method description is precise.
- Promising results in the fully binarized setting, which is difficult to obtain
- One of the few papers attempting binarization for natural language generation
- Great scaling law study of binary models on a large training set - which is perhaps the most interesting part of the paper

**Weaknesses:**

- Binarized activation results are still quite poor.
- Main results only test on one benchmark (WMT17 en-de)

Although it is contemporary, please consider comparing to the following work:
https://arxiv.org/abs/2306.01841
Having some strong established baseline could make the conclusions of the work stronger.

**Questions:**

none

**Limitations:**

Yes

---

> ### Author Rebuttal · Authors · 2023-08-09
>
> **Q1: Binarized activation results are still poor.**
>
> Yes, as highlighted in line 237 in Section 4.1, one challenge we identified in this work is, more precisely, that the attention layer activations are the bottleneck to a high-quality binarized Transformer for a sequence generation task. We hope our analysis will shed light on the future improvement to it.
>
> **Q2: Main results only test on one benchmark (WMT17 en-de)**
>
> Despite the WMT17 en-de dataset, we also train the model using our in-house translation corpus that contains 3-billion web-crawled sentence pairs in Section 4.2 and evaluate on both in-domain and out-of-domain datasets that span multiple categories. The detailed train and evaluation dataset information is provided in Appendix A.1. More evaluation results can be found in Appendix Figure 7, 8, and 9. In the main paper we put the key results and distill the knowledge we learnt.
>
> **Q3: Although it is contemporary, please consider comparing to the suggested work.**
>
> Thanks for sharing this very valuable contemporary work. Since it uses a different dataset than ours, we need to re-evaluate its approach and compare. Though we cannot finish the comparison during rebuttal given our dataset size, we will definitely cite it in related works.

---

### Official Review · Reviewer_HM3D · 2023-07-06

**Soundness:** 3 good
**Presentation:** 3 good
**Contribution:** 3 good
**Rating:** 5
**Confidence:** 4

**Summary:**

The authors introduce a new technique for binarization in Transformers that can be applied to machine translation known as Binarized Neural Machine Translation (BMT). They have adapted the binarization functions and training methods from PokeBNN to help address the "inflated dot-product variance" issues that arise when using one-bit weights and activations. The authors propose the use of LayerNorm in place of fixed scaling factors and make some architectural changes to improve the quality of the binarized model. Experiments show the BMT have the ability to scale and generalize effectively in both in-domain and out-of-domain settings.



**Strengths:**

1. The analysis of Variance Inflation in Binarization in Section 3.2 is interesting. The BERT model should also have this problem. What is the difference between it and the Transformer structure?
2. The paper provides a comprehensive experimental evaluation of the proposed BMT model on a 3-billion in-house parallel corpus. The authors make detailed analysis on the scaling law study and demonstrate the binary models can achieve the same BLEU score as float models with a smaller size.

**Weaknesses:**

1. Pre-LayerNormalization Transformer has already been proposed for a few years, and what's the difference with the Section 3.4 Replacement of Scaling Factor with LayerNorm?
"On Layer Normalization in the Transformer Architecture"
2. There is a lack of comparison with other quantization methods for Transformer models.
3. Based on the experimental results of in-house training data in Section 4.3, BMT still has about 2 BLEU gap compared to the floating-point model.

**Questions:**

1. The authors claim that "Experiments on the WMT dataset show that a one-bit weight-only Transformer can achieve the same quality as a float one, while being 16× smaller in size." Could you please clarify how the model size reduction by a factor of 16 is defined here? Is it just a reduction in storage size of the model weights?
2. In lines 57-60, the authors mention that "each word in the output translation sequence affects the generation of the next word", so what is the quality impact of binary quantization on long text generation?





**Limitations:**

The authors did not state any relevant limitations of the method.

---

> ### Author Rebuttal · Authors · 2023-08-09
>
> **Q1: What’s the difference between the proposed layernorm in Section 3.4 compared to the existing pre-layernorm Transformer?**
>
> Note that pre- or post-layernorm for the Transformer architecture indicates the layernorm position **outside** of the entire FFN module. Whereas in our proposal, each linear layer **inside** the FFN module needs to be followed by another layernorm (scaling factor). They are extra layernorms and only needed for 1-bit models to address the dot-product variance inflation.
>
> **Q2: There is a lack of comparison with other quantization methods for Transformer models.**
>
> We compared other binarization methods for Transformer in Table 1, last row (labeled as “Base”). We outlined the literature on transformer quantization in Section 2 beginning line 79. Many previous works focused on 8-bit and 4-bit quantization. Only a few of them focused on binarization. Among the binarization works, the model is BERT and the binarization function is directly applied with additional distillation methods. We adopted their quantization method to our encoder-decoder Transformer and listed the result in Table 1 in the last row.
>
> **Q3: Based on the experimental results of in-house training data in Section 4.3, BMT still has about 2 BLEU gap compared to the floating-point model.**
>
> Yes, that's correct. We concluded that in Section 4.3 line 332. There is still room to improve BMT when the dataset is comprehensive. One way is, as the scaling law suggests, scaling up the binarized model. As stated in line 333, the 30L6L binary model achieves the same BLEU score as the 8L6L float model while being still 6.7x smaller in model size.
>
> **Q4: Could you please clarify how the model size reduction by a factor of 16 is defined? Is it just a reduction in storage size of the model weights?**
>
> Yes, the model size reduction is defined by the reduction in the amount of bits used to store the weights, which is very important for model serving as highlighted in the challenges in Section 1 from line 24. Currently, model weights are typically stored as float16 or bfloat16 for efficiency reasons. Our technique allows the storage as 1 bit per weight, achieving a 16x compression. We will clarify this in the paper as well, thank you.
>
> **Q5: What is the quality impact of binary quantization on long text generation?**
>
> We analyzed the impact of binarization on translation generation quality in Section 4.3. Many of the translation sentences have dozens of tokens and some even longer. Some samples can be seen in Appendix B.1. We synthesized the knowledge learnt from there: (1) there will be around 2 BLEU quality loss at the same model size; (2) the quality loss can be recovered by scaling up the binary model a bit or generating more samples when selecting the translation text.
>
> Additionally, we study scaling law behavior on several in-house and open-source datasets (Appendix, Table 2). These datasets are a mix of different sequence lengths, ranging from short to long range sequences. From Figure 7 and 8 (appendix), we don't observe any outstanding difference in the slopes (p_e, p_d) of the scaling law on any dataset. This denotes that binary and float models have a similar scaling behavior in various scenarios with different sequence lengths, domains, naturalness etc.
>
> **Q6: The authors did not state relevant limitations of the method.**
>
> We will make the limitations more clear. (1) As stated in line 237, the activation-activation matmul binarization quality is still not ideal. We list 8 BMT variants in Table 1, among which BMT-8 (the one with activation-activation matmul binarization) has the largest loss and BLEU drop. We imagine a lot of the future effort will be dedicated to it. (2) As stated in the last part of the conclusion, there are several unanswered questions in this work, for example, shall we scale up the depth or width for the binarized Transformer and how should we design a mixed-precision scheme using binarization and potentially other formats? We will expand these discussions in the next revision.

---

> > ### Comment · Reviewer_HM3D · 2023-08-17
> > **After rebutal**
> >
> > Thanks for your answer.

---

### Official Review · Reviewer_jHYb · 2023-07-07

**Soundness:** 3 good
**Presentation:** 3 good
**Contribution:** 3 good
**Rating:** 6
**Confidence:** 4

**Summary:**

This paper presents a binarized neural translation model based on an encoder-decoder structure. The proposed method initially analyzes the challenges associated with binarized encoder-decoder models. The primary challenges arise from the significant impact of binarizing both weights and activations on result variance. Therefore, the authors primarily employ two methods to control variance: incorporating a scaling weight and adding a layer normalization layer.

To assess the effectiveness of the method, the authors report promising results obtained from experiments. The results indicate that the bottleneck stems from the attention activations. Binarizing different parts demonstrates a considerable variance in performance. However, with careful consideration of the binarization position, the proposed model achieves competitive results while significantly reducing its size. The ablation study further demonstrates the effectiveness of the proposed method in mitigating variance.

**Strengths:**

The motivation behind the method is clearly defined. The method begins by analyzing the reasons behind the failure of directly binarizing the weights. It was discovered that this failure can be attributed to a variance problem where binarization statistically inflates the magnitude, resulting in abnormal signal propagation within the neural network. To address this issue, the authors implemented two widely-used solutions: scaling weight and layer normalization. By employing these approaches, the authors were able to develop a binarized machine translation model that yields competitive results.

Furthermore, the authors conducted scaling law experiments, revealing that binarized models also exhibit scaling law characteristics. Despite the impact of binarization on model performance, the gap can be bridged by increasing the model size. Due to the significant reduction in model size achieved through binarization, the proposed model can effectively deliver superior performance with a smaller model size.

**Weaknesses:**

The main idea contribution of this work revolves around identifying and addressing the challenge of variance in binarizing machine translation models. However, the utilization of layer normalization and scaling weights in a straightforward manner somewhat limits its novelty and originality.


It appears that there is a lack of comparison with 8-bit or 4-bit results. Are these methods also subject to the scaling law, where larger models yield better performance?

**Questions:**

In Table 1, it is unclear which row represents the result of the BMT model with a size of 25MB. Does it imply that all rows correspond to models of approximately 25MB? Each row applies binarization to different weights and activations, resulting in significant performance variance across various settings.

---

> ### Author Rebuttal · Authors · 2023-08-09
>
> **Q1: Are 8-bit or 4-bit subject to the scaling law?**
>
> Thanks for pointing out this comparison and we will add it to the scaling law section. 8-bit and 4-bit models are studied more often and they do exhibit a scaling law where larger models yield better performance [1]. However, in the previous study the scaling law breaks starting from 3 bits where models seem not to converge. In our study we are able to re-establish the scaling law even for 1-bit models. We show that the dot-product variance is the key and a simple scaling factor can be a remedy. We also show in the ablation section 5 that indeed without a scaling factor a 1-bit model cannot converge but it can if with.
>
> [1] Tim D. et al., The Case for 4-bit Precision: k-bit Inference Scaling Laws, arxiv 2023.
>
> **Q2: In Table 1, it is unclear which BMT variant has a size of 25MB. Does it imply that all rows correspond to models of approximately 25MB?**
>
> As indicated by the caption, models with 1-bit weights will have 25MB of size. It means a model whose W_QKV, W_out, W_FFN are all labeled by checkmarks, indicating they are binarized, has a model size of 25MB. In Table 1, all BMT variants except for BMT-2 have 25MB. BMT-2 has only FFN binarized, which is also a potentially useful special variant that we want to demonstrate.

---

> > ### Comment · Reviewer_jHYb · 2023-08-20
> >
> > Thanks for the kind response. I would like to keep my score.

---

### Official Review · Reviewer_arnv · 2023-07-08

**Soundness:** 2 fair
**Presentation:** 2 fair
**Contribution:** 2 fair
**Rating:** 6
**Confidence:** 4

**Summary:**

This work proposes to binarize matrix multiplication for significantly saving memory and, thus, reducing latencies at inference time that is crucial for serving encoder-decoder model. Basic idea is to employ a binary variant of weights and inputs with scaling parameters in feed-forward and multi-head attention computation. Since the scaling hyparameters are critical for the binarized model, this work employs layer normalization to alleviate the issue so that appropriate normalization is performed automatically. Experiments are carried out mainly for WMT en-de by comparing against float variants with different number of parameters.

**Strengths:**

* Although the idea of binarizing matrix multiplication is now new, this work has a couple of contributions to binarize an encoder-decoder architecture. For example, the use of layer normalization sounds good to me given the stability analysis for scaling hyperparameters.

* Binarization is performed not only feed-foward, but also multi-head attention to reduce the computation. For stability, residual connection is introduced in the model following the prior work, but the choice is carefully designed.

* Experiments are systematically carried out by varying the number of parameters and the proposed method is compared with the float variants.

**Weaknesses:**

* Although this work has comparisons in terms of loss and translation qualities measured by, e.g., BLEU, this work is not presenting actual speed measured by seconds. I understand the condition might be varied, it is better to run experiments to see whether the proposed method is actually faster than a float baseline.

* Discussion in residual connection is a bit weak, and only a figure is presented. I feel better to show by equations to avoid any confusions. Also, further analysis on why residual connection is needed in the output projection will be a plus for this submission.

* Given the binarization, it might be prone to high variance in experimental results. It would be good to run some analysis by running multiple times and show averages/variances.

* Translation qualities are measured only by BLEU. Given the limitations of the metric, better to present alternative metrics as well, e.g., BLEURT or COMET.

* No experiments for larger data. It is minor, though, better to run larger data to see if the proposed method will also work or not.

**Questions:**

* There exists almost no clear description about how binarization is applied in section 3.5. Given Figure 1, it sounds like layer norm is shared across query, key and value, and it is not clear what is the motivation of sharing.

**Limitations:**

Better to discuss the current limitation on the experiments, e.g., scale and variances.

---

> ### Author Rebuttal · Authors · 2023-08-09
>
> **Q1: I understand the condition might be varied, it is better to measure the speedup of the proposed method.**
>
> Thank you for commenting on the speedup measurement. It is not accessible because we are not aware of an ecosystem (accelerator combined with software stack) that supports such measurement. However, there is convincing evidence that 1-bit matmuls will have high performance. For example, NVIDIA’s A100 architecture [1] shows 1-bit matmul is 8x faster than 8-bit measured by TFLOPS throughput, though requiring NVIDIA’s own assembly. Also [2] shows binary matmul is 9x-12x faster compared to 8-bit measured by latency on ARM CPUs. In that light, among other goals, our work aims to contribute to the assessment of whether binary ML hardware+software is a viable future direction for the ML accelerator industry.
>
> [1] NVIDIA Ampere Architecture Whitepaper. Table 3.
>
> [2] Tom B., et al., Larq Compute Engine: Design, Benchmark, and Deploy State-of-the-Art Binarized Neural Networks, MLSys’21.
>
> **Q2: Better to show equations on the residual connection and an analysis on why it is needed in the output projection.**
>
> Thanks for the suggestion. We will add an equation on residual connection in Section 3.5: Out = LN(X*W) + X, where X is the output of score-value einsum. As briefly discussed in Section 3.5, we added the residual connection because the binarization of the output projection layer will possibly mislead the optimizer since the gradients through a binarization function are computed by the straight-through estimator. The shortcut link will carry the raw gradient from the previous layer, therefore partially mitigate this issue. We also conducted an ablation study in Section 5(b) to demonstrate the effectiveness of the proposed shortcut.
>
> **Q3: Would be good to show averages/variances of results.**
>
> We were able to reproduce the model loss ourselves within several runs, but since we need to train each model size for one million steps on three billion sentence pairs and we have limited hardware resources, the number of losses for each result is not sufficient to establish statistical analysis. However, given the dataset and model size, we expect that the variance on experimental results should be small. We will attempt to add such analysis in the final version.
>
> **Q4: Better to present alternative metrics as well, e.g., BLEURT or COMET.**
>
> We measured and reported BLEURT in Section 4.3. In that section we compared the model generation quality using both BLEURT and BLEU on our in-house translation dataset. We show that BLEURT scores also improve as we scale up the model size. Additionally, we run MBR decoding with BLEURT metric and show that given enough MBR samples, binary models match the quality of float models. We will highlight it better in the paper.
>
> **Q5: No experiments for larger data. It is minor, though, better to run larger data to see if the proposed method will also work or not.**
>
> The scaling law study in Section 4.2 and model generation quality study in Section 4.3 are both carried out on our in-house large production-scale translation dataset. This dataset has 3 billion En-De sentence pairs and is one of the largest translation datasets in the ML community.
>
> **Q6: How binarization is applied in Section 3.5? Why is layernorm shared across QKV in Figure 1?**
>
> We will clarify this in the paper. Binarization will be applied via casting the inputs right before a matmul to 1-bit using the function defined in line 106. Naming it as x_b= bin(x) for short, a binarized linear layer A*W will be computed as bin(A) * bin(W).
> Each of the QKV projections has its own independent layernorm, i.e., the parameters in layernorm are not shared. Figure 1 draws a big rectangle to represent the layernorms because in practical implementations the QKV projections are usually combined into a large single matmul. Thanks for pointing it out, and we will add corresponding captions to reflect this.
>
> **Q7: Better to discuss the current limitation on the experiments.**
>
> We will make the limitations more clear. (1) As stated in line 237, the activation-activation matmul binarization quality is still not ideal. We list 8 BMT variants in Table 1, among which BMT-8 (the one with activation-activation matmul binarization) has the largest loss and BLEU drop. We imagine a lot of the future effort will be dedicated to it. (2) As stated in the last part of the conclusion, there are several unanswered questions in this work, for example, shall we scale up the depth or width for the binarized Transformer and how should we design a mixed-precision scheme using binarization and potentially other formats? We will expand these discussions in the next revision.

---

> > ### Comment · Reviewer_arnv · 2023-08-15
> > **After rebuttal**
> >
> > Thanks for your answers.
> >
> > - Q5: I think I missed the large scale experiments.
> >
> > - Q4: If f my understanding is correct, Figure 4(b) is reporting BLEURT score for MBR decoding. Basically, the figures are comparing two different decoding strategies, i.e., beam search by searching for the best translation according to the model, and MBR by taking consensus in the sampled translations. It is not clear why showing BLEU for beam search and BLEURT for MBR, which is leading to non-systematic comparison. I would suggest the comparison should be systematic, e.g., showing both BLEU and model-based score, e.g., BLEURT and Comet, for both decoding strategies to see whether the trends are the same. Also note that MBR-BLEURT will be heavily biased toward BLEURT given that BLEURT is employed for MBR. I would suggest a different metric, Coment, for a fair comparison.

---

> > > ### Author Response · Authors · 2023-08-16
> > > **Response to follow-up comments**
> > >
> > > Thanks for commenting on Q4 and providing very good suggestions.
> > >
> > > **“I would suggest showing both BLEU and model-based scores for both decoding strategies to see whether the trends are the same.”**
> > >
> > > Presenting a Cartesian product of (MBR, beam search) x (BLEU, BLEURT/COMET) is indeed a good suggestion and ablation. We will attempt such measurement. The goal of Figure 4 is to compare whether binary and float models produce translations of similar qualities (or how large is the gap). Therefore, we initially chose a common setting of beam search decoding + BLEU score. Despite that, we were also aware of the discussion about the limitations of BLEU in the machine translation community. We thus provided the quality evaluation in another common setting of MBR + BLEURT, and we wanted to show that the lessons concluded in Section 4.3 are independent of the two settings.
> > >
> > > **“MBR-BLEURT will be heavily biased toward BLEURT given that BLEURT is employed for MBR. I would suggest a different metric, Comet, for a fair comparison.”**
> > >
> > > It is indeed true that MBR decoding with metric-XYZ will be biased towards metric-XYZ, but we also observed that the machine translation community adopts MBR+BLEURT for two reasons:
> > > - MBR+BLEURT correlates the most with Oracle Human MQM evaluation (Table 2, 3, last column [1]).
> > > - BLEURT is one of the most accurate evaluation metrics (Table 2, [2]).
> > >
> > > Our motivation was to compare binary vs float models on “one” of the most adopted model-based setups in translation research, so we chose this combination as part of the evaluation.
> > >
> > > We also agree that showing one more neural metric (eg. COMET) would be a more comprehensive evaluation and we will attempt to do so in our final revision.
> > >
> > > [1] Markus Freitag et al., High Quality Rather than High Model Probability: Minimum Bayes Risk Decoding with Neural Metrics. ACL’22.
> > >
> > > [2] Tom Kocmi et al., Large Language Models Are State-of-the-Art Evaluators of Translation Quality. EAMT’23.

---

### Author Rebuttal · Authors · 2023-08-09

We thank all reviewers for their positive feedback, considering our problem analysis and empirical experiments as a good contribution to the community. We also appreciate all comments and suggestions. We will address questions below in separate threads.

---

### Decision · Program_Chairs · 2023-09-21

**Decision:**

Accept (poster)

**Comment:**

The paper proposes a novel quantization scheme to binarize transformer machine translation models. The method includes inserting additional layer normalization for activations and additional residual connections. The authors demonstrate good results on the WMT test set especially for weight-only binarization, and promising results for both weight and activation binarization.

After the rebuttal, the concerns of reviewers are well-addressed.